# How Does Low-Carbon Financial Policy Affect Corporate Green Innovation?—Re-Examination of Institutional Characteristics, Influence Mechanisms, and Local Government Behavior

Hailin Yao [1] , Zeyi Wan [1] , Huixiang Zeng [1] and Qingfang Wu [2],*

1 Business School, Central South University, Changsha 410083, China; yaohailin14703@163.com (H.Y.); lillian021023@icloud.com (Z.W.); zenghx1120@csu.edu.cn (H.Z.)
2 Department of Planning and Finance, Central South University, Changsha 410083, China
* Correspondence: 212032@csu.edu.cn

**Abstract:** This paper employs a propensity score matching approach to construct a control group and estimate the impact of the CETS pilot policy, a low-carbon financial policy, on corporate green innovation and its impact mechanism in a difference-in-difference manner. The results show that the CETS pilot policy has a significantly positive effect on corporate green innovation. The higher the penalty degree and the carbon price, the more obvious the promotion of the green innovation of pilot enterprises. The mechanism test shows that the improvement of corporate green innovation is mainly due to the incentive effect rather than the anti-driving effect of the CETS pilot policy, that is, the policy promotes corporate green innovation by providing innovation resources and enhancing the willingness to innovate. Further analysis shows that only in regions where local governments have less competitive pressure can the CETS pilot policy effectively promote enterprise innovation resources and that a close and clean government–business relationship can help strengthen the promotion effect of the CETS pilot policy on the willingness of enterprises to innovate. Furthermore, this paper introduces its theoretical framework as a strategic tripod to explore the friction in the process of the CETS pilot policy affecting corporate green innovation from the perspective of the industry environment and corporate resources. This research shows that a lack of industry green technology and corporate human capital may hinder the positive impact of the CETS pilot policy on corporate green innovation. Finally, this study found that the CETS pilot policy has no significant impact on the quality of corporate green innovation, and the lack of industry green technology and corporate human capital may hinder the CETS pilot policy from improving the quality of corporate green innovation.

**Keywords:** carbon emission trading system; green innovation; carbon price; incentive effect; local government behavior

## 1. Introduction

Currently, the concept of a green and low-carbon economy has acquired wide recognition in the international market, and international trade competition has focused on green and low-carbon qualities and added value. China urgently needs to examine this new trend to seize the competitive high ground. In September 2020, at the 75th United Nations General Assembly, China solemnly proposed the aim of achieving its "carbon peak by 2030 and carbon neutrality by 2060". Green technology innovation will be the core driving force and vital guarantee for achieving the goal of "carbon neutrality". Green innovation refers to technological innovation that reduces environmental contamination and reduces the use of raw materials and energy [1,2]. In the short term, handling the contradiction between economic transformation and carbon constraints urgently requires green technology's support; in the long term, enhancing China's competitiveness in the international low-carbon market depends on green technology innovation. However, compared with

traditional technological innovation, green technological innovation has the characteristics of a large upfront investment, extended return period, and difficult-to-predict risks, so it must be supplemented by long-term and large-scale financial support [3,4].

Carbon emissions trading systems are considered to have advantages such as technological incentives, cost-effectiveness, and emission reduction efficiency [2]. Essentially, they are a financial instrument that guides funds towards low-carbon sectors. Carbon emissions trading refers to the government allocating carbon emission quotas to various emitting entities based on their emission reduction targets and allowing these entities to freely trade their quotas. Therefore, developing the carbon market and encouraging carbon trading will help guide funds towards green and low-carbon technology companies.

Internationally, the European Union (EU) is the leading economic entity that first priced carbon emissions and adopted market-based trading, leading the development of the global carbon market. The EU Carbon Market (EU ETS) strictly implements the Cap and Trade system. EU member countries need to specify detailed allocation plans (NAP), listing the names of emission control companies and emission reduction targets. After review, emission quotas (EUA) will be allocated to various sectors and companies. The United States has not yet established a national carbon emission trading system, but there are regional emission reduction plans that have been established by various states, mainly including the Regional Greenhouse Gas Initiative (RGGI), the Western Climate Initiative (WCI), and the Transportation and Climate Initiative Program (TCI-P). In addition, countries such as Australia, New Zealand, Japan, and South Korea have also established carbon markets to varying degrees. Under positive factors such as the implementation of a stable reserve mechanism in 2019 and the return of the Green Party, the EU carbon market has accelerated the reduction of its carbon quotas and the carbon price has grown rapidly. The United States, as one of the pioneers of carbon emissions trading, is currently in a state of coexisting regional carbon markets due to the lack of a unified trading system. The carbon market in East Asia was initiated by South Korea, and the construction of carbon emissions trading systems in China and Japan is gradually accelerating. The "2023 Global Emissions Trading Status Report" by the International Carbon Action Partnership (ICAP) found that the international carbon market remains stable in the face of soaring energy costs. With the increase in economic pressure and the impact of the global energy crisis, governments are more committed to reducing our dependence on fossil fuels and playing a key role in the carbon emissions market.

In October 2011, the National Development and Reform Commission of China issued a notice about launching carbon emission trading system (CETS) pilots. Seven provinces and cities including Beijing, Shanghai, Tianjin, Chongqing, Hubei, Guangdong, and Shenzhen were approved to carry out CETS pilots. Subsequently, the Chinese government implemented CETS pilots in seven pilot provinces and cities from 2013 onwards. The national carbon emission trading market officially began trading on 16 July 2021. As significant participants in the carbon trading market, enterprises are both major carbon emitters and core organizations in the development of low-carbon products [5]. Green innovation is one of the most effective methods for enterprises to achieve carbon reduction and benefit from the carbon trading market. Therefore, it remains to be tested whether the pilot policy of carbon emission trading, as an important means to achieve China's "carbon neutrality" goal, can effectively promote the green innovation of enterprises.

At present, there are relatively few studies on the relationship between carbon emission trading and enterprises' green innovation in China, and there are some scholars that have different views. Scholars who support the promotion theory believe that carbon emission trading has an "innovation compensation" effect, which can prompt enterprises to actively carry out green technology innovation activities to reduce pollution costs and improve their competitiveness [6,7]. Raza, Z. [8] believes that appropriate environmental regulatory policies can encourage companies to innovate in terms of green technology. The profits from innovation can partially or fully cover the costs of environmental management, creating a compensatory effect for innovation. Giulio et al. [9] used data from EU companies to explore

the factors influencing companies' development and application of green innovation. The results show that environmental policies can prompt companies to carry out green innovation. Hu Jun et al. [10] found that China's carbon emission trading has a positive impact on the number of enterprise patents filed but did not further explore its path of action. Scholars who support the inhibition theory believe that the pilot policy of carbon emission trading will force enterprises to occupy more production and innovation resources to achieve pollution reduction and carbon reduction in order to meet policy standards. It cannot promote the green innovation of enterprises and may even inhibit the green innovation of enterprises [2]. Zhang et al. [11] found that carbon emission trading has a crowding-out effect on corporate R&D investment, which increases the price of carbon trading and thus inhibits corporate green innovation. Zhang et al. [12] found that the impact of China's carbon trading policy on the technological innovation of pilot enterprises showed obvious industry heterogeneity. That is to say, carbon trading helps to improve the technological innovation of power and aviation enterprises but has no significant impact on six other heavily polluting industries. Chen et al. [2] found that China's carbon trading pilot policy significantly inhibited enterprises' green innovation. Enterprises chose to reduce production in the short term to reduce carbon emissions. The reduction of cash flow and expected income prompted enterprises to cut investment in R&D activities and inhibited green innovation. The existing literature has the following limitations: first, there is a lack of understanding of the characteristics of carbon trading policies, and the intrinsic logic of carbon trading and green innovation has not yet been deeply deconstructed from the point of view of institutional design and market characteristics; secondly, the path of action of the carbon trading pilot policy on enterprises' green innovation is still unclear; finally, there is a lack of attention to local governments as policy implementers when discussing the effects of carbon trading policies. Therefore, a deep investigation of our country's carbon emissions trading, especially its core system design and market characteristics, in terms of the incentive methods and mechanisms of corporate green innovation activities, helps us to deeply understand the theoretical mechanisms of the carbon reduction policy tools inducing green innovation and more fully elucidate the key role of the carbon trading system, an important policy tool, in the green and sustainable development of our economy.

Based on the above considerations, this article selects listed companies in the Shanghai and Shenzhen A-shares from 2009 to 2019 as its sample, uses the PSM-DID model, and empirically tests the implementation effect of the carbon emissions trading pilot policy implemented in China since 2013, that is, whether the quality of corporate green innovation has been effectively improved. At the same time, this article will deeply analyze the effect of carbon prices and punishment systems on the carbon trading market.

The marginal contributions of our study are as follows: First, in terms of research perspective, our study focuses on the financial instrument characteristics of carbon trading and deeply analyzes the institutional design and market characteristics of policies, that is, analyzes the impact of the punishment intensity of pilot policies and carbon market prices on green innovation, so as to more comprehensively evaluate the effects of carbon emission trading pilot policies and expand the existing research on carbon trading. Secondly, regarding its path of action, we examine both the incentive effects of innovation resources and innovation willingness and the reverse effect of environmental costs. We also analyze the path of action of carbon trading policies in terms of government behavior, revealing the mechanism by which carbon trading policies affect corporate green innovation and showing the close connection between carbon trading policies and local government behavior in their implementation and effectiveness. Furthermore, in analyzing the policy effectiveness boundaries, this paper analyzes the friction in the level of green technology in the industry and the human capital of enterprises based on a strategic tripod theory framework [13]. The strategic tripod analysis framework is proposed based on the institutional background of emerging economies, and it is believed that institutional factors usually determine a strategy together with industry and enterprise factors [14]. This analysis framework is applicable to emerging economies such as China, and by examining corporate government dependence

and industry regulation as moderating variables, it expands the boundary conditions of meaningful innovation, matches the development environment of our enterprises well, and provides a comprehensive analysis perspective [14]. Finally, the research results of our study may provide a reference for policy makers in China to improve the national carbon emission trading system.

The rest of this article is arranged as follows: the second part is the Theoretical Analysis and Hypothesis Development; the third part is the Research Design; The fourth part is the Analysis of Empirical Results; the fifth part is a Path Analysis; the sixth part is a Heterogeneity Analysis; the seventh part is an Extended Analysis; and the eighth part is the conclusion and policy recommendations.

## 2. Theoretical Analysis and Hypothesis Development

### 2.1. The Impact of the Carbon Emission Trading Pilot Policy on Enterprise Green Innovation

We predict that China's carbon emission trading pilot policy will positively affect the green innovation of pilot enterprises for the following two reasons.

First, carbon trading policies are essentially financial instruments that redirect funds to low-carbon fields and provide expected revenue incentives for pilot enterprises to encourage green technology innovation. Since corporate managers generally believe that energy conservation, emission reduction, and pollution control are costly, and the risk and cost of innovation failure are relatively high, they are less enthusiastic about investing in green innovation [4]. When facing internalized emission reduction costs, carbon trading pilot enterprises need to choose the "optimal" emission and R&D level and balance the costs and benefits, making the market return of innovative activities particularly important. However, environmental policies help to increase the expected returns of green innovation, correct environmental externalities, and thus stimulate the green innovation activities of companies [15,16]. On the one hand, in the carbon trading market, enterprises with more advanced emission reduction technologies can benefit from selling carbon quotas to enterprises with relatively higher emission reduction costs, which helps enhance their motivation to invest in pollution reduction with green innovation as the focus. On the other hand, the carbon trading market uses carbon quotas as commodities, transmitting value signals for energy conservation and emission reduction through carbon prices. This can lower the concerns and raise the enthusiasm for the green R&D of corporations. Generally, the carbon trading policy has increased corporate expectations for green innovation revenue while weakening concerns about green innovation risks. This increases the focus on and promotion of green innovation and technology among pilot enterprises.

Second, the carbon trading policy will also bring cost pressure to pilot enterprises and force them to carry out green technology innovation. According to institutional theory, in order to improve their organizational legitimacy, enterprises must comply with the various policies and regulations established by the government. In order to comply with carbon trading policies, pilot enterprises can achieve pollution and carbon reduction by purchasing carbon quotas, reducing their production, and investing in green technology research and development to achieve legitimacy. However, according to the induced innovation theory proposed by Hicks [17], the main purpose of corporate technological innovation is to reduce the use of production factors with relatively high costs. Therefore, the direction of corporate technological innovation will be influenced by changes in the relative prices of its factors. The purchase of carbon quotas and production cuts will damage corporate profits and are not conducive to the sustainable development of enterprises. The emission trading system causes the price of carbon emissions rights, which are considered factors, to rise, relatively speaking. In order to save scarce factors (i.e., carbon emission rights), companies intend to control their carbon dioxide emissions by developing new technologies. At this time, companies are willing to pay the cost of green innovation to reduce their long-term emission reduction costs. Therefore, driven by the principles of profit maximization and sustainable development, enterprises may turn to green technology research and development to alleviate the environmental protection pressure they face. At the same time, carbon trading

essentially internalizes pollution externalities. Due to the large differences in emission control capability and energy utilization efficiency between enterprises, enterprises with poor emission control capabilities and relatively low energy utilization efficiency will be at a disadvantage in the carbon trading market and face more severe competitive pressure. This will force enterprises to pay attention to energy conservation and emission reduction and, therefore, force them to carry out green technology innovation activities.

In summary, carbon emission trading pilot policies bring revenue incentives and cost pressures to pilot enterprises, which helps enterprises carry out green innovation. Therefore, the first hypothesis is proposed as follows:

**H1:** *The carbon emission trading pilot policy will have a positive impact on the green innovation of pilot enterprises.*

The punishment mechanism and carbon price, respectively, reflect the government administrative control and market incentive mechanisms of the carbon trading policy [18] and are the two major elements for the orderly and efficient operation of the carbon trading policy. All carbon trading pilot areas have established a punishment system for emission control enterprises that have not fulfilled their quota-clearing obligations. The punishment system mainly includes two methods: rectification within a time limit and fines. In addition, some pilot areas have also added supporting punishment measures such as an inclusion in credit records and restrictions on funding to urge enterprises to strictly fulfill their relevant obligations in terms of their quota management. The punishment system is all at the firm level, that is, the size of the penalty and other means of punishment will be determined according to the degree of the firm's violation. Existing research has found that environmental penalties not only cause economic losses to enterprises, but also negatively affect the reactions of consumers and outside investors, thereby leading to a decline in corporate market value [19,20]. Therefore, the punishment mechanism will increase the legitimacy pressure on enterprises, ensure that enterprises meet the environmental performance standards proposed by the policy [21], and urge enterprises to reflect on their own shortcomings in green development and overcome the organizational inertia that does not want to change. Specifically, punishment measures such as fines and the cancellation of preferential policies in the carbon trading market will increase the cost pressure on pilot enterprises. Being included in the credit system will not only directly hinder the external financing of enterprises, but may also damage their corporate image, lead to negative market expectations, and adversely affect their long-term development. Wu Yinyin et al. [18] found that, when comparing different punishment measures in various pilot areas, the pilot areas with stronger punishment intensity have a greater warning effect on enterprises and better regional carbon reduction effects. Therefore, we expect that the greater the punishment intensity, the greater the legitimacy pressure perceived by pilot enterprises, the greater the risk of future financial and market value losses, and the more the punishment can promote enterprises to carry out green technology transformation and upgrading and reduce their dependence on conventional, polluting production methods.

The carbon price signal is the fundamental characteristic and core function of the carbon market, reflecting short-term emission reduction costs, as well as the value of green technology, and affecting the stability and effectiveness of the carbon trading market. Bu Wenke and Zhao Mengen [22] found that carbon prices can play a role in resource allocation, positively affect the stock prices of new energy enterprises, have a negative impact on the stock prices of traditional energy enterprises in the short term, and show a positive impact in the long term. This article argues that the price signal of the carbon market has a dual function in affecting enterprises' green innovation. On the one hand, the carbon price signal reflects emission reduction costs and forces enterprises with higher marginal emission reduction costs to carry out green transformation and green technology innovation; on the other hand, the carbon price signal reflects the value of carbon emissions and encourages enterprises with lower marginal emission reduction costs to adopt green

and low-carbon technology in their production and operation and pay more attention to green technology in R&D activities [23], thereby benefiting from selling their remaining carbon quotas. Therefore, we expect that as the carbon market price rises, the greater the cost pressure and economic incentives perceived by pilot enterprises and the more enterprises are motivated to carry out green technology innovation.

In summary, as the carbon emission trading policy becomes more stringent, the higher the carbon price of the carbon market, the greater the environmental protection pressure and economic incentives perceived by pilot enterprises, and the more likely pilot enterprises are to pursue green innovation. Thus, the following hypotheses are proposed:

**H2a:** *The greater the punishment of the carbon trading pilot, the more obvious the role of green innovation promotion in pilot enterprises.*

**H2b:** *The higher the carbon price of the carbon trading market, the more obvious the role of green innovation promotion in pilot enterprises.*

*2.2. The Path of the Impact of the Carbon Emission Trading Pilot Policy on Enterprise Green Innovation*

The carbon emission trading policy has both flexible characteristics and certain mandatory features. Therefore, we predict that carbon trading may affect enterprise green innovation through incentive effects or force effects.

(1) Incentive effect: innovative resources.

The carbon emission trading pilot's management policies stipulate relevant green subsidy measures. We expect that the carbon trading pilot will encourage pilot enterprises to carry out green innovation by supplementing their innovative resources due to two factors. Firstly, based on a resource-based view, the carbon trading pilot can raise capital for enterprises with green subsidies, reducing their innovation costs and thus promoting enterprise green innovation. Green innovation requires long-term and large-scale investment, but under short-term performance and cash flow pressure, managers are often forced to abandon high-investment and high-risk green innovation [24]. The green subsidies that are stipulated by the carbon emission trading policy can provide financial support for the green innovation activities of enterprises, alleviate financing constraints, reduce the marginal cost of enterprises' own green innovation efforts, and disperse the risks of enterprises' green innovation activities, thereby promoting green innovation activities.

Secondly, according to signaling theory, green subsidies act as a signal of favorable investment that is transmitted to outside investors. This can help enterprises obtain external innovative resources to improve their green innovation performance. Green innovation is generally considered to bring more risks and uncertainties than other innovative activities [25]. Receiving green government subsidies can signal government recognition of enterprises and their green R&D activities. This will increase the confidence of outside investors in their green innovation activities and help them provide more financing and other green innovation resources for enterprises. This leads to the following hypothesis:

**H3a:** *The carbon emission trading pilot policy promotes the green innovation of pilot enterprises by supplementing their innovative resources.*

(2) Incentive effect: innovation willingness.

The theory of planned behavior posits that the behavior of a subject is driven by the subject's willingness [26]. The carbon emission trading pilot policy can directly affect the green innovation behavior of pilot enterprises to a certain extent, but, in most cases, it will not directly lead to green innovation behavior. Wang and Zhang [27] believe that willingness is the intermediary and bridge between environmental regulation and behavior. To a certain extent, enterprises can stimulate green innovation behavior only when

they have the willingness to innovate and when they accumulate and integrate resources and knowledge.

The carbon trading pilot policy has the characteristic of flexibility. The carbon trading policy uses carbon prices as a signal to convey the value of carbon assets. This allows enterprises to freely trade carbon quotas in the carbon trading market, which helps enterprises generate and strengthen an internal motivation and subjective desire for technological innovation driven by profit-seeking, that is, innovation willingness. Innovation willingness reflects how much enterprises accept and recognize technological innovation behavior [27]. The stronger the innovation willingness of pilot enterprises, the more likely they are to invest more resources and energy and more actively implement green innovation practices [28] to adapt to the carbon emission trading policy and seize the potential opportunities and economic returns brought by green innovation strategies. In other words, pilot enterprises included in carbon trading will tend to regard green innovation to protect the environment as a development opportunity and increase their enthusiasm for new ideas and new paths. This will improve their innovation willingness and promote their green innovation. Therefore, we hypothesize that the carbon emission trading pilot policy will promote green innovation by improving the innovation willingness of pilot enterprises. This leads to the following hypothesis:

**H3b:** *The carbon emission trading pilot policy promotes the green innovation of pilot enterprises by improving their innovation willingness.*

(3) Reverse effect: environmental cost.

The carbon trading pilot policy has certain mandatory characteristics. The carbon trading pilot incorporates enterprises with large carbon emissions into its quota management and uses a default penalty system to constrain enterprises' carbon emissions and ensure effective market operation. Enterprises whose carbon emissions exceed their carbon quota need to purchase emission quotas on the carbon market to pay for their environmental costs. This will cause pilot enterprises to pay attention to the current situation of environmental pollution, actively reflect on their own shortcomings, and break through the constraints of original organizational conventions. In addition, green development has become a realistic demand from external stakeholders towards heavily polluting enterprises [24]. The public is increasingly concerned about environmental issues and pursuing green products. Xu et al. [29] found that outside investors often give lower valuations to enterprises that have been penalized for environmental protection while giving higher valuations to enterprises that pursue green development. However, the disclosure of enterprises' pollution information in China is low and remains difficult to obtain. The public and outside investors often cannot assess the true pollution situation of enterprises. All seven carbon emission trading pilots in China use the quota of the carbon emission rights of enterprises as a selection criterion for inclusion. Therefore, being included in the carbon emission trading market may send a signal to the public and outside investors that the enterprises have poor pollution control in their production activities, which is not conducive to the image and reputation of enterprises and will increase the external environmental pressure faced by enterprises. Green innovation can enhance the public's confidence in the green development of enterprises, reduce their negative expectations of pollution emissions in their production and operation, and improve the reputation and value of enterprises [24]. Therefore, the negative signal of "heavy pollution" generated by the carbon emission trading policy will increase the environmental pressure on enterprises, induce them to increase their environmental governance costs, and then force them to adjust their environmental strategies and carry out green technology innovation activities to improve their social reputation and competitiveness. Therefore, the following hypothesis is proposed:

**H3c:** *The carbon emission trading pilot policy forces the green innovation of pilot enterprises by increasing environmental costs.*

### 2.3. Friction in the Processes of the Carbon Trading Pilot Policy Affects the Green Innovation of Enterprises

Carbon trading may not have an immediate positive impact on the green technology innovation of pilot enterprises. There may be friction in the process of carbon trading, which prevents carbon trading from fully playing its role in promoting enterprise green innovation in the short term. The limitations and challenges that may exist in the process of carbon trading promoting the green innovation of pilot enterprises are mainly confined to two aspects. On the one hand, green technology innovation is difficult and risky, and the requirements of related its technical foundations are correspondingly high. On the other hand, the promotion of enterprise green innovation through a carbon trading policy cannot be separated from the input of a series of complementary resources. Carbon trading is essentially a carbon financial instrument that can guide funds to green and low-carbon fields, but enterprises also need other inputs to align, including low-carbon talent training, carbon asset management, and the adaptation of production and operation activities.

Peng et al. [13] integrated an institution-based view, industry-based view, and resource-based view to construct a strategic tripod theoretical framework to answer questions about enterprise strategy and performance. According to their strategic tripod theoretical framework, the strategy and performance of a company are jointly influenced by various factors at the levels of the institution, industry environment, and enterprise resources [13,14]. Therefore, based on a strategic tripod theoretical framework, this article attempts to use the level of industry green technology as an industry condition and human capital as a corporate resource to explore the impact of the two types of friction on the effectiveness of carbon trading policies. Figure 1 shows the research framework of this article.

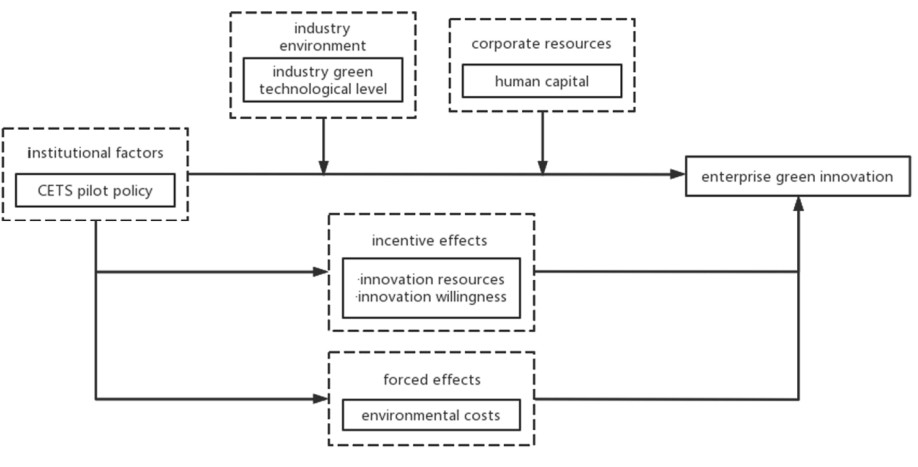

**Figure 1.** Research framework.

Industry green technology level. The industry green technology level refers to the degree of overall green technology's research and development and application in an industry. From a technical support perspective, the development of green technology in an industry is an important guarantee for pilot companies to achieve green technology innovation.

First, the higher the industry green technology level, the more it can provide pilot companies with a better innovation environment and technical foundation. For pilot companies, green technology within the same industry has a wider potential applicability and can provide general solutions for enterprises encountering heterogeneous technical problems [30], which helps companies acquire and apply new knowledge to sustain their



green technology innovation. In addition, a higher industry green technology level means that the industry has developed and applied more extensive and influential technologies for energy conservation and emission reduction, which helps pilot companies redeploy existing and familiar industry knowledge and establish connections with new knowledge, reducing the risk and uncertainty of their green technology innovation activities. Therefore, the higher the industry green technology level, the more it can enable carbon trading pilot enterprises to use existing resources efficiently and drive them to carry out green innovation activities under the expected income incentives of the carbon market. On the other hand, if the level of industry green technology is too low, it will bring obstacles in the green technology innovation of pilot enterprises.

Secondly, the higher the level of green technology in an industry, the more mimetic pressure it can exert on pilot enterprises. Generally speaking, the level of green technology in an industry reflects the adoption of green technological innovation by a company's peers. The more peers that adopt green innovation, the more mimetic pressure it exerts on companies to adopt similar green innovation practices [31]. For example, when senior managers of a company lack the knowledge to maintain their legitimacy, their safest strategy is to imitate their successful peers [32,33]. Therefore, when the level of green technology in an industry is high, pilot enterprises will tend to imitate their peers and actively engage in green innovation to respond to carbon trading policies and achieve legitimacy. On the other hand, if the level of technology in an industry is too low, it may trigger a "convergence effect", weakening the positive impact of carbon trading policies on corporate green innovation.

Human capital. High-quality talent is mainly reflected in the years of education people have received and the advantages they have in their technical skills, innovation ability, and their comprehensive quality [34]. From the perspective of talent, a company's participation in the carbon trading market and its green technological innovation cannot be separated from high-quality talent. On the one hand, high-quality talent can help pilot enterprises accurately understand the carbon trading market system and provide support for their carbon asset management decisions. They can also convey the strategic importance of green production to senior executives of the enterprise, which is conducive to the implementation of green innovation practices. At the same time, high-quality talent can identify the market opportunities and resources that may brought about by carbon trading and help enterprises seize opportunities for green innovation and profit. On the other hand, talent is the main force of enterprises' green innovation. Both human capital theory and new growth theory suggest that human capital is an important driving factor of innovation [35]. High-quality talent with rich knowledge reserves has advantages in acquiring, integrating, and applying knowledge. The interaction of the technical knowledge and thinking methods of internal talent can stimulate innovative ideas and provide intellectual support for the success of green innovation. On the contrary, if an enterprise lacks high-quality talent reserves, they will not be conducive to the pilot enterprises' understanding of carbon trading policies and market opportunities. This will also hinder their practice of green low-carbon concepts and the application of green knowledge within the enterprise, thus bringing obstacles to enterprises' green technological innovation.

Therefore, this article further purposes the following hypotheses:

**H4a:** *The effect of the pilot policy of carbon emission trading on the green innovation improvement of pilot enterprises will be more obvious in samples with a higher green technology level in their industry.*

**H4b:** *The effect of the pilot policy of carbon emission trading on the green innovation improvement of pilot enterprises will be more obvious in samples with a higher level of human capital.*

### 3. Research Design

#### 3.1. Model Construction

Our study uses the Propensity Score Matching Difference-in-Differences (PSM-DID) model to test the impact of the pilot policy of carbon emission trading on enterprises' green innovation. Since 2013, the Chinese government has successively launched carbon emission trading systems in seven provinces and cities, its pilot areas, including Beijing, Guangdong, Shanghai, Shenzhen, Tianjin, Chongqing, and Hubei. Referring to the practice of Quan et al. [36], our study constructs two dummy variables, PILOT and POST, and then establishes the following difference-in-differences model to obtain samples through a propensity score matching method to test H1, that is, the impact of carbon emission trading on pilot enterprises' green innovation.

$$GreenPat_{i,t} = \alpha_0 + \alpha_1 PILOT_i \times POST_t + \alpha_2 PILOT_i + \alpha_3 POST_t + \alpha_4 CV_{i,t} + \mu_i + \lambda_i + \gamma_t + \varepsilon_{i,t} \tag{1}$$

The subscripts i and t represent the enterprise and time, respectively. $GreenPat_{i,t}$ represents the green innovation of enterprise i in year t. $PILOT_i$ is a dummy variable for whether the enterprise is included in carbon emission trading systems. When enterprise i is included in carbon emission trading systems during the sample period, $PILOT_i$ is assigned a value of 1, otherwise it is 0. $POST_t$ is a time dummy variable. After the launch of the CETS pilot policy, that is, when $t \geq 2013$, $POST_t$ is assigned a value of 1, otherwise it is 0. Our study designs a difference-in-differences model (1) to observe the regression coefficient $\alpha_1$, which is the net effect of the CETS pilot policy on enterprises' green innovation after excluding other influencing factors. If the coefficient $\alpha_1$ in model (1) is significantly positive, it indicates that the CETS pilot policy has a significant positive impact on enterprises' green innovation. $CV_{i,t}$ represents a set of control variables. Finally, our study controls for the fixed effects of province ($\mu_i$), industry ($\lambda_i$), and year ($\gamma_t$). $\varepsilon_{i,t}$ is a random disturbance term.

At the same time, in order to verify H2a and H2b, we construct models (2) and (3):

$$GreenPat_{i,t} = \alpha_0 + \alpha_1 Penalty_{i,t} + \alpha_4 CV_{i,t} + \mu_i + \lambda_i + \gamma_t + \varepsilon_{i,t} \tag{2}$$

$$GreenPat_{i,t} = \alpha_0 + \alpha_1 Price_{i,t} + \alpha_4 CV_{i,t} + \mu_i + \lambda_i + \gamma_t + \varepsilon_{i,t} \tag{3}$$

where $Penalty_{i,t}$ represents the penalty intensity of the emission trading pilot policy for defaulting enterprises. In Formula (2), if $\alpha_1$ is significantly positive, it indicates that the greater the penalty intensity of the carbon trading pilot policy, the greater the promotion effect on the green innovation of enterprises. $Price_{i,t}$ is the carbon price in the carbon emission trading market. In Formula (3), if $\alpha_1$ is significantly positive, it indicates that the higher the carbon price, the greater the promotion effect on the green innovation of enterprises.

#### 3.2. Sample Selection and Data Processing

To avoid the impact of the 2008 financial crisis and the 2020 major public health event on business operations, our study selected Shanghai and Shenzhen A-share listed companies from 2009 to 2019 as the research sample. We performed the following processing steps: (1) excluding enterprises in the financial insurance industry; (2) excluding samples with missing data for the main regression variables; (3) excluding ST and *ST enterprises; and (4) considering that matching needed to be performed in the period before the launch of the carbon trading pilot, we excluded enterprises that were listed in 2013 or later. Our study manually collects and collates the list of listed companies that are included in the carbon trading pilot and the related penalty measures implemented from 2013 to 2019 through the official websites of the Development and Reform Commissions of the seven pilot provinces and cities. Patent application data come from the China National Intellectual Property Administration, and other data such as corporate finance and carbon market transaction prices come from the CSMAR database.

*3.3. Variable Definition*

(1) Dependent variable

The dependent variable in our study is the green innovation of enterprises (GreenPat). Following the practice of Li and Xiao [24], we use a logarithm of one plus the number of green patents applied for by the enterprise in that year as a measure of innovation. Specifically, our study uses the IPC code to identify green patents based on the green technology list of the World Intellectual Property Organization (WIPO).

(2) Independent variables

The independent variables in our study are PILOT and POST, which are dummy variables representing policy and time, respectively. In addition, when analyzing the characteristics of carbon trading, we use two variables: carbon price (Price) and penalty intensity (Penalty). Following the practice of Wu et al. [18], we use the logarithm of the annual average of daily closing prices to measure carbon prices. In addition, following the practice of Wu et al. [18], we conduct a comprehensive evaluation of the penalty intensities in the seven pilot areas based on the content of various penalty systems such as the penalty amount and supporting penalty regulations. Finally, the ranking of penalty intensity in each pilot carbon market is as follows: Beijing, Shanghai, and Shenzhen are tied for first place, Chongqing is second, Hubei third, Guangdong fourth, and Tianjin fifth. Therefore, our study assigns scores to each pilot market based on the ranking of their penalty intensity. The penalty intensity of Beijing, Shanghai, and Shenzhen's carbon markets is 5, that of Chongqing's carbon market is 4, Hubei's carbon market is 3, Guangdong's carbon market is 2, and Tianjin's carbon market is 1.

(3) Control variables

Following the practices of Quan et al. [36] and Ma et al. [37], our study selects the following control variables: enterprise size (SIZE), represented by the logarithm of total assets; enterprise age (AGE), represented by the logarithm of the observation year minus the year of establishment plus 1; return on assets (ROA), represented by the net profit divided by total assets; asset–liability ratio (LEV), represented by total liabilities divided by total assets; cash flow level (CFO), represented by the net cash flow from operating activities divided by total assets; management incentives (SHARE), represented by management shareholding divided by total share capital; and equity concentration (COCEN), represented by the shareholding ratio of the largest shareholder.

## 4. Analysis of Empirical Results

*4.1. PSM Method Analysis*

The enterprises included in the carbon market are not randomly selected, so there might be selection bias in the treatment effect. Therefore, before testing the policy effect with the DID model, we used the propensity score matching method (PSM) to find enterprises not included in the carbon market with similar characteristics to form a control group for the enterprises included in the carbon market to obtain more accurate estimates. Firstly, following the practice of Wang et al. [35], the characteristic variables of the treatment group and control group samples in the year before the carbon emission trading pilot policy (i.e., 2012) are selected as matching data, including enterprise size (SIZE), enterprise age (AGE), asset–liability ratio (LEV), return on assets (ROA), and cash flow level (CFO). Secondly, a Logit model is used to estimate the propensity score and conduct 1:1 nearest neighbor matching with a caliper of 0.05. Lastly, balance tests and propensity score kernel density tests are performed on the matching samples to ensure the reliability of the matching results. For balance tests, Rosenbaum and Rubin [38] believe that if the standard deviation after matching is less than 20%, the matching effect is good. According to the results in Table 1, compared with those before matching, all matching variables have reduced differences between the treatment group and control group after matching, and the absolute value of the standard deviation range is less than 7% after matching. Moreover, all t-statistics are not significant, indicating that there is no significant difference between the treatment group and control group after matching. Therefore, the quality of matching

in our study is good. Figure 2 illustrates the distribution of propensity scores in the treatment group and control group before and after matching. It can be seen that the propensity scores of the treatment group and control group are more similar after matching, and their distribution is almost overlapping. This also indicates that the quality of the matching is good. After matching, our study's research sample consists of 2197 enterprise-year observations from a total of 201 enterprises, including 106 carbon-market-included enterprises and 104 non-carbon-market-included enterprises.

**Table 1.** Balance test results.

| Variable Name | | Average | | Standard Deviation Range (%) | Standard Deviation Reduction Range (%) | T-Test | |
|---|---|---|---|---|---|---|---|
| | | Treatment Group | Control Group | | | t | p > ItI |
| SIZE | Before matching | 22.675 | 21.815 | 57.3 | 95.5 | 7.06 | 0.000 |
| | After matching | 22.54 | 22.579 | −2.6 | | −0.18 | 0.856 |
| AGE | Before matching | 2.6109 | 2.6226 | −2.9 | −97.0 | −0.30 | 0.764 |
| | After matching | 2.6148 | 2.6378 | −5.8 | | −0.43 | 0.668 |
| LEV | Before matching | 0.43978 | 0.39708 | 20.3 | 95.5 | 2.03 | 0.042 |
| | After matching | 0.43954 | 0.4376 | 0.9 | | 0.06 | 0.950 |
| ROA | Before matching | 0.0471 | 0.04589 | 2.6 | 81.6 | 0.27 | 0.787 |
| | After matching | 0.04625 | 0.04647 | −0.5 | | −0.04 | 0.971 |
| CFO | Before matching | 0.06589 | 0.04469 | 32.4 | 78.8 | 3.01 | 0.003 |
| | After matching | 0.06421 | 0.05972 | 6.9 | | 0.53 | 0.598 |

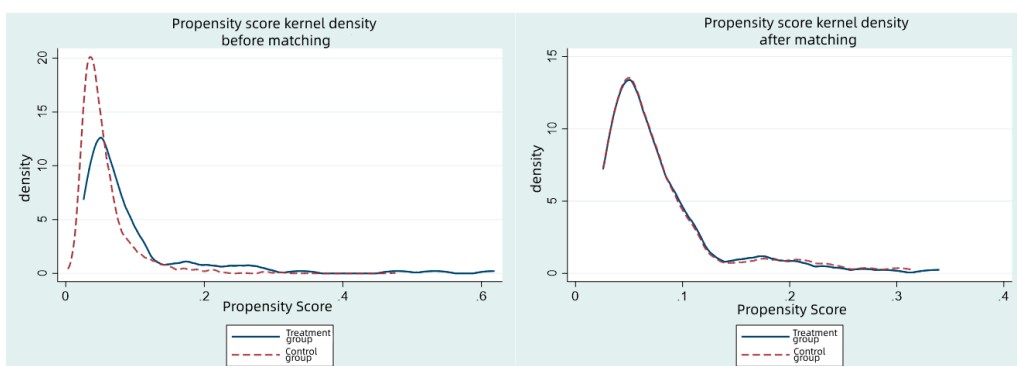

**Figure 2.** Distribution of propensity scores before and after matching.

*4.2. Descriptive Statistics and Significance Tests*

Panel A of Table 2 reports the descriptive statistical results of the main variables after matching. Among them, the data on penalty intensity and carbon price only include pilot data from 2013 to 2019. According to Panel A of Table 2, the mean value of green technology innovation in the observed sample is 0.5493, indicating that the research enterprises are not very active in terms of green technology innovation activities. Panel B of Table 2 is the result of an inter-group significance test of the dependent variable. It can find out whether there is a significant difference in the number of green patent applications between enterprises included in carbon emissions trading and those not. This preliminarily test verifies that the carbon emission trading pilot policy does have a significant impact on enterprise green innovation.

**Table 2.** Descriptive statistics and difference tests.

| | Observation | Average | Median | Standard deviation | Minimum | Maximum |
|---|---|---|---|---|---|---|
| **Panel A: Descriptive Statistics** | | | | | | |
| GreenPat | 2197 | 0.5493 | 0 | 1.0183 | 0 | 6.4938 |
| PILOT | 2197 | 0.5043 | 1 | 0.5001 | 0 | 1 |
| POST | 2197 | 0.6650 | 1 | 0.4721 | 0 | 1 |
| PILOT×POST | 2197 | 0.3359 | 0 | 0.4724 | 0 | 1 |
| Penalty | 723 | 4.6030 | 5 | 0.9139 | 1 | 5 |
| Price | 723 | 3.5353 | 3.5786 | 0.5226 | 1.4096 | 4.3681 |
| SIZE | 2197 | 22.7911 | 22.5860 | 1.5188 | 19.6837 | 28.3412 |
| AGE | 2197 | 2.7748 | 2.8332 | 0.4173 | 0.6931 | 4.7875 |
| LEV | 2197 | 0.4514 | 0.4637 | 0.2133 | 0.0075 | 0.9248 |
| COCEN | 2197 | 0.3854 | 0.3801 | 0.1607 | 0.0641 | 0.9900 |
| SHARE | 2197 | 0.1017 | 0.0003 | 0.1895 | 0 | 0.8972 |
| ROA | 2197 | 0.0458 | 0.0384 | 0.0545 | −0.6243 | 0.3739 |
| CFO | 2197 | 0.0578 | 0.0561 | 0.0697 | −0.2941 | 0.3773 |

| | **Non-CETS pilot policy-included enterprises** | | | **CETS pilot policy-included enterprises** | | | Average difference | Median difference |
|---|---|---|---|---|---|---|---|---|
| **Panel B: Average–Median difference test** | | | | | | | | |
| | Sample size | Average | Median | Sample size | Average | Median | | |
| GreenPat | 1089 | 0.3573 | 0 | 1108 | 0.7380 | 0 | −0.3807 *** | 65.5924 *** |

Note: The values in parentheses are t-values, and *, **, and *** indicate that the statistics are significant at the 10%, 5%, and 1% levels, respectively.

### 4.3. Baseline Regression and Feature Analysis

Columns (1) and (2) of Table 3 report the results of the difference-in-differences regression of the carbon trading pilot policy and enterprise green innovation. According to the empirical test results in column (1) and (2), the POST×PILOT coefficient is significantly positive at the 5% level. With the economic meaning of its regression coefficient, and keeping other conditions unchanged, the number of green patent applications of enterprises included in the carbon emissions trading pilot policy increased by 0.1558, and the number of green patent applications of enterprises not included in the carbon emissions trading pilot policy increased by 0.1534. This indicates that, while controlling other influencing factors, the green innovation output of enterprises included in the carbon emissions trading pilot policy is higher. That is, the carbon emission trading pilot policy can promote enterprise green innovation, so H1 is verified.

**Table 3.** The impact and characteristic analysis results of the CETS pilot policy.

| Variables | GreenPat | | | |
|---|---|---|---|---|
| | (1) | (2) | (3) | (4) |
| PILOT×POST | 0.1534 ** | 0.1558 ** | | |
| | (2.0650) | (2.2254) | | |
| POST | 0.1638 * | −0.2350 ** | | |
| | (1.7902) | (−2.4779) | | |
| PILOT | 0.2668 *** | 0.3829 *** | | |
| | (3.7075) | (5.5109) | | |
| Price | | | 0.1707 * | |
| | | | (1.8883) | |
| Penalty | | | | 0.4033 *** |
| | | | | (5.9314) |
| SIZE | | 0.2707 *** | 0.3926 *** | 0.3398 *** |
| | | (9.1626) | (6.3567) | (5.5095) |
| AGE | | 0.0797 | 0.0614 | 0.2309 * |
| | | (1.3900) | (0.5611) | (1.9519) |
| LEV | | 0.4091 *** | 0.4900 * | 0.6184 ** |
| | | (3.1552) | (1.8256) | (2.3421) |
| ROA | | 0.6148 * | 0.4342 | 0.6040 |
| | | (1.8022) | (0.6940) | (1.0137) |
| CFO | | 0.3249 | 0.7423 | 0.7000 |
| | | (1.1951) | (1.0695) | (1.0212) |
| SHARE | | 0.2473 ** | 0.1973 | 0.0326 |
| | | (2.2287) | (0.8668) | (0.1422) |

**Table 3.** *Cont.*

| Variables | GreenPat | | | |
|---|---|---|---|---|
| | **(1)** | **(2)** | **(3)** | **(4)** |
| COCEN | | −0.7920 *** | −1.0233 *** | −0.9688 *** |
| | | (−5.7819) | (−3.8869) | (−3.7538) |
| Constant | −0.2678 ** | −6.5754 *** | −8.6103 *** | −9.2558 *** |
| | (−2.2758) | (−10.0457) | (−6.3084) | (−7.2085) |
| Industry fixed effects | YES | YES | YES | YES |
| Province fixed effects | YES | YES | YES | YES |
| Time fixed effects | YES | YES | YES | YES |
| N | 2197 | 2197 | 723 | 723 |
| $R^2$ | 0.3533 | 0.4484 | 0.5442 | 0.5593 |

Note: The values in parentheses are t-values, and *, **, and *** indicate that the statistics are significant at the 10%, 5%, and 1% levels, respectively.

The regression results of the feature analysis of the carbon emission trading pilot policy are shown in columns (3) and (4) of Table 3. The results show that the regression coefficient of the penalty intensity of the carbon trading pilot policy is significantly positive at the 1% level, indicating that there is a positive correlation between the penalty intensity of the carbon trading pilot policy and the number of green patent applications made by enterprises. With each increase in the level of the punishment intensity of the carbon trading pilot policy, the number of green patent applications made by enterprises increases by 0.4033. This means that the greater the penalty intensity of the carbon trading pilot policy, the more obvious the effect IS of promoting green innovation in pilot enterprises, and H2a is verified. The regression coefficient of the carbon price is significantly positive at the 10% level, indicating that the higher the carbon price in the carbon trading market, the more obvious the effect is of promoting green innovation in pilot enterprises, and H2b is verified.

*4.4. Robustness Test*

In order to ensure the reliability of the conclusions of our study, a series of robustness tests were carried out, as follows:

(1). Parallel trend test. The use of a difference-in-differences model for analysis requires the assumption of parallel trends, that is, in the absence of the implementation of the carbon emission trading pilot policy, the development trend of enterprise green innovation in the treatment group and the control group would be consistent. Our study tests whether the common trend assumption of the DID model holds, that is, whether the treatment group samples and control group samples have similar characteristics and trends before the implementation of the carbon emission trading policy, so as to further test the reliability of the regression results. Following the approach of Huang and Qi [39], we use an event study method to conduct a parallel trend test with the following specific model settings:

$$GreenPat_{i,t} = \alpha_0 + \sum_{k=-4}^{6} \beta_k D_{i,t}^k + \mu_i + \lambda_i + \gamma_t + \varepsilon_{i,t} \tag{4}$$

where D is a set of dummy variables, $D_{i,t}^k$ (k = −4, −3, −2, −1) represents the treatment group in the Kth year before the implementation of the carbon trading policy at time t; and $D_{i,t}^k$ (k = 1–6) represents the treatment group i in the kth year after the implementation of the carbon trading policy at time t. Our study takes the second year before the implementation of the carbon emission trading pilot policy (2011) as the base period.

The estimated values of the coefficients for each period before the implementation of the carbon emission trading pilot policy can be used to test whether the trends of the treatment group and control group samples were parallel before the former was included in the carbon trading policy. The estimated values of the coefficients for each period thereafter can be used to describe the distribution of green innovation incentive effects each year

after the implementation of the carbon emission trading pilot policy. As shown in Figure 3, the hollow circles in the figure represent the size of the regression coefficients that depict the effect of the carbon emission trading pilot policy, and the dashed line represents the confidence interval (the confidence interval is 95%). Our study finds that the regression coefficients of explanatory variables fluctuate gently around 0 before the implementation of the carbon emission trading pilot policy, indicating that there is no significant difference in green patent applications between the experimental and control groups, satisfying the parallel trend assumption.

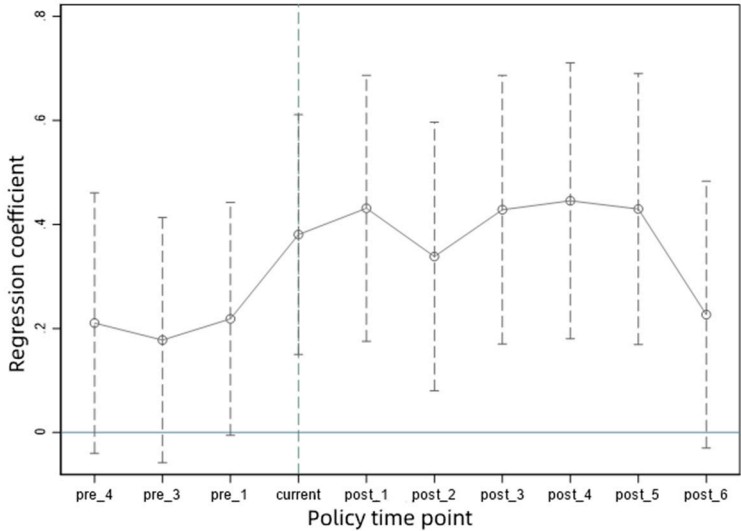

**Figure 3.** Parallel trends test.

(2). Alternative dependent variables. Other unobservable factors that affect the green patent applications of enterprises may interfere with our research conclusions. Following the approach of Xu and Cui [3], our study uses the ratio of green patent applications to total patent applications for robustness testing. This indicator helps to further eliminate confounding factors that simultaneously affect green patent applications and total patent applications. In order to ensure the robustness of our benchmark regression results, the dependent variable in model (1) is replaced with the green patent application ratio (RatioGreenPat) to test the effect of the carbon emission trading pilot policy on the ratio of green patent applications. The regression results are shown in Table 4, column 1. From the regression results, it can be seen that the POST×PILOT coefficient is significantly positive at the 1% level. This indicates that after eliminating the confounding factors that may simultaneously induce enterprises to apply for green patents and all patents, the effect on green innovation of the carbon emission trading pilot policy is still significant. This corroborates the robustness of the above benchmark analysis; that is, the carbon emission trading pilot policy does promote enterprise green innovation.

**Table 4.** Partial robustness test results.

| Variables | Alternative Dependent Variables RatioGreenPat (1) | Winsorizing Test GreenPat (2) | Shortening the Event Window Period GreenPat (3) |
|---|---|---|---|
| PILOT×POST | 0.0302 *** | 0.1591 ** | 0.1627 ** |
| | (2.6125) | (2.3367) | (2.0820) |
| PILOT | 0.0184 | 0.3664 *** | 0.3540 *** |
| | (1.6149) | (5.3902) | (4.7904) |
| POST | 0.0399 ** | −0.2099 ** | −0.1347 |
| | (2.5583) | (−2.2620) | (−1.4612) |

**Table 4.** *Cont.*

| Variables | Alternative Dependent Variables RatioGreenPat (1) | Winsorizing Test GreenPat (2) | Shortening the Event Window Period GreenPat (3) |
|---|---|---|---|
| SIZE | −0.0066 * | 0.2444 *** | 0.2612 *** |
| | (−1.8532) | (9.0862) | (7.6723) |
| AGE | −0.0193 * | 0.0730 | 0.0613 |
| | (−1.7374) | (1.1785) | (0.9659) |
| LEV | 0.0675 *** | 0.4247 *** | 0.4845 *** |
| | (2.9004) | (3.2533) | (3.1615) |
| ROA | −0.0322 | 0.4864 | 0.0773 |
| | (−0.5376) | (1.1203) | (0.1494) |
| CFO | 0.0290 | 0.3010 | 0.2342 |
| | (0.5601) | (1.0612) | (0.7174) |
| SHARE | 0.0356 * | 0.2173 ** | 0.3159 ** |
| | (1.8910) | (1.9750) | (2.4825) |
| COCEN | −0.0065 | −0.6992 *** | −0.7216 *** |
| | (−0.2780) | (−5.3793) | (−4.6160) |
| Constant | 0.1581 * | −6.0146 *** | −6.3882 *** |
| | (1.8468) | (−10.0333) | (−8.5077) |
| Industry fixed effects | YES | YES | YES |
| Province fixed effects | YES | YES | YES |
| Time fixed effects | YES | YES | YES |
| N | 2197 | 2197 | 1573 |
| $R^2$ | 0.229 | 0.445 | 0.441 |

Note: The values in parentheses are t-values, and *, **, and *** indicate that the statistics are significant at the 10%, 5%, and 1% levels, respectively.

(3). Winsorizing test. Considering that the outliers in the enterprise data may affect the regression results of our study, our study follows the approach of Chen et al. [40] and winsorizes the continuous variables in the 1st and 99th percentiles to ensure the robustness of the model's conclusions. From the regression results in column (2) of Table 4, it can be seen that after performing a two-sided trimming of the relevant continuous variables in the 1% quantile, the POST×PILOT coefficient is still significantly positive at the 5% level, which that also corroborates the robustness of the above benchmark analysis.

(4). Shortening the event window period. National carbon emission trading officially started in China in 2017. In order to eliminate the interference of the national carbon trading market, our study follows the approach of Li et al. [41] and conducts a regression analysis on samples from 2009 to 2016. From the regression results in column (3) of Table 4, it can be seen that the POST×PILOT coefficient is significantly positive at the 5% level, which also supports the conclusion that "the carbon emission trading pilot policy has a significant positive impact on the enterprise green innovation".

## 5. Path Analysis

### 5.1. Incentive Effect Test

In order to test whether an incentive effect of innovation resources and innovation willingness exists, our study constructs two indicators: innovation resources (Subsidy) and innovation willingness (Intention). For them, our study used green subsidies as a proxy variable for innovation resources, measured by the logarithm of the annual government green subsidies received by an enterprise plus 1. Specifically, following the approach of Hu et al. [42], we identified and manually organized subsidy projects and amounts with keywords such as "environmental protection", "green", "low carbon", "energy saving", and "emission reduction", based on the government subsidy details disclosed in the annual report of the enterprise, and then summed them up as the annual government green subsidy amount received by the enterprise. In addition, following the approach of Jiang and Liu [43],

we used enterprise R&D investment as a measure of innovation willingness (see details in Appendix A). R&D investment is measured by the ratio of R&D investment to operating income. The data come from the CSMAR and Wind databases. The specific test results are shown in Table 5. Here, columns (1) and (2) tested the mediating role of innovation resources. It can be seen that, in the impact of the carbon emission trading pilot policy, innovation resources have a partial mediating effect on enterprise green innovation, thus verifying H3a. This shows that pilot enterprises effectively use the green subsidies from the carbon emission trading policy for green R&D activities and obtain external resources through the positive signals from green subsidies to achieve more green innovation output. Columns (3) and (4) tested innovation willingness. It can be seen that, in the impact of the carbon emission trading pilot policy, innovation willingness does have a partial mediating effect on enterprise green innovation, that is, H3b is verified. This shows that the carbon emission trading policy can enable pilot enterprises to invest more R&D resources and actively carry out green innovation under the drive of profit-seeking.

**Table 5.** Path analysis results.

| Variables | Incentive Effects: Green Subsidies | | Incentive Effects: Innovation Willingness | | Forced Effects: Environmental Costs | |
|---|---|---|---|---|---|---|
| | Subsidy (1) | GreenPat (2) | Intention (3) | GreenPat (4) | PPE (5) | Expense (6) |
| PILOT×POST | 1.5565 *** | 0.1455 ** | 0.0051 ** | 0.1345 * | 0.0094 | −0.0010 |
| | (2.9350) | (2.0873) | (2.4466) | (1.9319) | (0.9092) | (−0.2618) |
| Subsidy | | 0.0066 * | | | | |
| | | (1.9528) | | | | |
| Intention | | | | 4.2121 *** | | |
| | | | | (3.7850) | | |
| PILOT | −1.2194 ** | 0.3910 *** | −0.0012 | 0.3882 *** | 0.0297 *** | −0.0144 *** |
| | (−2.3285) | (5.6090) | (−0.5849) | (5.6233) | (2.8060) | (−3.1047) |
| POST | −0.1862 | −0.2338 ** | 0.0208 *** | −0.3227 *** | −0.0496 *** | −0.0063 |
| | (−0.2549) | (−2.4697) | (6.8679) | (−3.2876) | (−3.2599) | (−1.1846) |
| SIZE | 0.4728 *** | 0.2676 *** | −0.0002 | 0.2717 *** | −0.0080 ** | −0.0095 *** |
| | (2.9537) | (8.9563) | (−0.3735) | (9.3044) | (−2.5024) | (−8.4167) |
| AGE | −0.2667 | 0.0814 | −0.0102 *** | 0.1227 ** | −0.0153 * | −0.0008 |
| | (−0.5910) | (1.4300) | (−5.1091) | (2.1316) | (−1.7345) | (−0.2138) |
| LEV | 0.9572 | 0.4028 *** | −0.0188 *** | 0.4884 *** | 0.0992 *** | −0.0210 ** |
| | (0.9413) | (3.1126) | (−4.8187) | (3.7730) | (4.8035) | (−2.2696) |
| ROA | −0.8774 | 0.6206 * | −0.0653** | 0.8897 *** | −0.4420 *** | −0.1584 *** |
| | (−0.3352) | (1.8207) | (−2.5720) | (2.5991) | (−5.9458) | (−4.7757) |
| CFO | 1.0579 | 0.3179 | −0.0020 | 0.3331 | 0.4249 *** | −0.0161 |
| | (0.5203) | (1.1687) | (−0.2341) | (1.2355) | (9.1928) | (−0.7666) |
| SHARE | −1.6953 ** | 0.2585 ** | 0.0134 *** | 0.1910 * | −0.1002 *** | −0.0032 |
| | (−2.1546) | (2.3376) | (3.0116) | (1.7756) | (−6.3307) | (−0.4376) |
| COCEN | −2.8534 *** | −0.7732 *** | −0.0012 | −0.7868 *** | −0.0907 *** | 0.0047 |
| | (−3.0740) | (−5.6174) | (−0.3240) | (−5.8145) | (−5.0596) | (0.4103) |
| Constant | −8.9397 ** | −6.5165 *** | 0.0497 *** | −6.7848 *** | 0.4036 *** | 0.2996 *** |
| | (−2.4861) | (−9.8444) | (3.3164) | (−10.3826) | (5.1632) | (9.6738) |
| Industry fixed effects | YES | YES | YES | YES | YES | YES |
| Province fixed effects | YES | YES | YES | YES | YES | YES |
| Time fixed effects | YES | YES | YES | YES | YES | YES |
| N | 2197 | 2197 | 2197 | 2197 | 2197 | 2197 |
| R$^2$ | 0.218 | 0.450 | 0.561 | 0.456 | 0.636 | 0.449 |

Note: The values in parentheses are t-values, and *, **, and *** indicate that the statistics are significant at the 10%, 5%, and 1% levels, respectively.

### 5.2. Forced Effect Test

This article also examines whether a path of action of the forced effect of carbon trading on enterprise green innovation exists. According to the research of Cui et al. [44], enterprises' environmental costs include capitalized environmental costs and expensed environmental costs.

Due to the large number of missing values in the data on these two indicators in the existing micro-enterprise database, referring to the practices of Xi and Zhou [45], this article uses the proportion of fixed assets to total assets (PPE) as a proxy variable for enterprises' capitalized environmental costs, and uses administrative expenses as a percentage of operating income (Expense) as a proxy variable for corporate expensed environmental costs. The data come from the CSMAR database. The specific test results are shown in columns (5) and (6) of Table 5. The regression coefficients of PILOT×POST are not significant, indicating that there is no significant correlation between the carbon emission trading policy and corporate environmental costs, thus indicating that a forced effect of the carbon emission trading pilot on enterprise green innovation is not established, that is, H3c has not been verified. This shows that, unlike punitive environmental regulations, carbon trading is essentially a financial tool for directing funds to low-carbon areas. It mainly uses the incentive effect of carbon asset values and supporting subsidies to encourage enterprises to carry out green innovation, rather than imposing cost pressures on enterprises.

### 5.3. Path Analysis of the Carbon Trading Pilot Policy from the Perspective of Government Behavioral Characteristics

In the context of fiscal decentralization in China, the behavior and decision-making of local governments can not only directly affect the allocation of fiscal resources, but also play an important role in the actual implementation of national policies. Although the previous text has conducted a detailed analysis of the implementation effect of the carbon trading pilot policy, it cannot fully reflect the role played by the pilot areas' governments as actual promoters of the carbon trading policy under the unified leadership of the central government. The carbon trading market is actually a market artificially established by the government, and the pilot area governments play a central role and are responsible for formulating, implementing, and supporting incentive systems; controlling participating entities; and maintaining order in the carbon trading market. Therefore, analyzing the impact of local government's behavioral characteristics is an important part of understanding and characterizing carbon trading pilot policies. First of all, fiscal subsidies are essentially an important reflection of the will and decision-making of local governments [46], that is, the green subsidies provided by carbon trading policies largely depend on the behavior and decision-making of local governments. At present, China's economy has shifted to a stage of high-quality development and no longer simply judges its success by its GDP growth rate; however, economic development remains one of the assessment criteria for officials. Therefore, how government competition behavior, under promotion incentives, affects the incentive effect of carbon trading pilot policies on enterprise innovation resources warrants further study. Secondly, whether the carbon emission trading policy can enhance the willingness of enterprises to innovate depends on their understanding of the potential opportunities and economic returns brought about by the pilot policy and their judgment of the innovation friendliness of their institutional environment. Local governments are precisely the main body implementing and communicating carbon trading pilot policies. Therefore, the interaction between government and enterprises and their communication mechanisms are very important. Therefore, the interactive and communicative mechanisms between the government and enterprises are very important. Therefore, an expanded analysis will be conducted on how local government competition behavior and government–business relations affect the mechanisms of carbon trading pilot policies.

In terms of local government competition behavior, based on the approach of Du et al. [47], this article measures government competition through the pursuit of higher economic levels in surrounding areas, using the highest GDP of neighboring provinces/GDP of the province where the enterprise is located. Based on the median, the enterprise samples are divided into two groups with high or low levels of government competition to estimate the incentive effect of the green subsidies from carbon trading policies. Government competition data come from the China Statistical Yearbook (see details in Appendix B).

In terms of government–business relations, since the 18th National Congress of the Communist Party of China, President Xi Jinping has repeatedly emphasized the building a collaborative and clean relationship between government and business. Drawing on the research of Zhou et al. [48] and Zhou et al. [49], this article measures government–business relations using the urban government–business relationship indicators developed by Nie et al. [50] and the closeness index and integrity index of government–business relations for each city. Here, closeness of the government–business relationship means how much the government communicates with enterprises, cares for and serves enterprises, and actively solves problems for enterprises. The cleanliness of the government–business relationship means how much the government is self-disciplined, maintains integrity, is transparent in its administrative law enforcement, and deals with enterprises in a legal and compliant manner. Therefore, based on the median of the closeness index and integrity index, the enterprise samples are split into two groups to estimate how the carbon trading policy affects the enhancement of innovation willingness.

The empirical results are shown in Table 6. Here, according to the results of columns (1) and (2), only if the local government has a low level of competition, meaning that it faces less competitive pressure, does the carbon emission trading pilot policy have a significant positive impact on the innovation resources of local enterprises. This indicates that, under low competitive pressure, local governments are better able to balance the goals of economic growth and low-carbon development, study and utilize carbon trading policies and their supporting incentive measures more effectively, and have enough resources to allocate fiscal funds flexibly to support the implementation of carbon trading policies and encourage the green development of pilot enterprises, thereby increasing the green subsidies available for enterprises through more effective carbon trading policies. In contrast, under high competitive pressure from local government, green innovation may suffer from managerial myopia and policy bias distortions prioritizing investment projects with quick results and evident growth effects and ignoring carbon trading markets with long payback periods and less obvious short-term economic performance enhancements. Moreover, local governments under high competitive pressure may also lack the capacity to coordinate the resources for various policies. They may fail to implement supporting incentive systems for carbon trading policies and may not offer strong support to enterprises.

**Table 6.** The heterogeneous impact of government behavior characteristics on the path of action of the CETS pilot policy.

| Variables | Subsidy High Level of Government Competition (1) | Subsidy Low Level of Government Competition (2) | Intention High Level of Closeness in Government–Business Relations (3) | Intention Low Level of Closeness in Government–Business Relations (4) | Intention High Level of Integrity in Government–Business Relations (5) | Intention Low Level of Integrity in Government–Business Relations (6) |
|---|---|---|---|---|---|---|
| PILOT×POST | 0.3254 | 2.4222 *** | 0.0064 ** | 0.0030 | 0.0059 * | 0.0020 |
| | (0.4000) | (3.0069) | (2.0061) | (0.9031) | (1.8805) | (0.6388) |
| POST | −0.7096 | 0.1191 | 0.0188 *** | 0.0181 *** | 0.0189 *** | 0.0210 *** |
| | (−0.6632) | (0.1036) | (3.7670) | (5.8004) | (3.7802) | (6.0308) |
| PILOT | 0.3526 | −1.4656 * | −0.0052 | −0.0007 | −0.0048 | 0.0108 *** |
| | (0.4043) | (−1.6752) | (−1.4583) | (−0.1680) | (−1.3508) | (2.7534) |
| SIZE | 0.7170 *** | 0.6501 *** | 0.0016 | −0.0006 | 0.0023 ** | −0.0007 |
| | (2.8786) | (2.6242) | (1.6442) | (−0.9002) | (2.2987) | (−0.8171) |
| AGE | −0.1738 | −0.9732 * | −0.0105 *** | −0.0070 ** | −0.0108 *** | −0.0111 *** |
| | (−0.2251) | (−1.6716) | (−4.2911) | (−2.4014) | (−4.4373) | (−3.7913) |
| LEV | 0.2398 | −1.4752 | −0.0303 *** | −0.0054 | −0.0251 *** | −0.0063 |
| | (0.1494) | (−0.9913) | (−4.8181) | (−1.0870) | (−4.1881) | (−1.1027) |
| ROA | −0.5210 | −2.1888 | −0.0688 * | −0.0625 *** | −0.0728 * | −0.0527 *** |
| | (−0.1172) | (−0.6079) | (−1.9226) | (−3.2723) | (−1.7845) | (−3.2985) |
| CFO | 2.2644 | 0.8430 | 0.0045 | 0.0029 | 0.0017 | −0.0009 |
| | (0.6659) | (0.3138) | (0.3642) | (0.3539) | (0.1342) | (−0.1113) |
| SHARE | −3.3436 ** | −1.1760 | 0.0195 *** | 0.0114 ** | 0.0255 *** | 0.0169 *** |
| | (−2.4949) | (−1.1008) | (2.7039) | (2.1626) | (3.5177) | (2.8661) |

**Table 6.** *Cont.*

| Variables | Subsidy<br><br>High Level of Government Competition<br><br>(1) | Subsidy<br><br>Low Level of Government Competition<br><br>(2) | Intention<br>High Level of Closeness in Government–Business Relations<br>(3) | Intention<br>Low Level of Closeness in Government–Business Relations<br>(4) | Intention<br>High Level of Integrity in Government–Business Relations<br>(5) | Intention<br>Low Level of Integrity in Government–Business Relations<br>(6) |
|---|---|---|---|---|---|---|
| COCEN | −3.4607 ** | −1.2028 | −0.0117 ** | 0.0182 *** | −0.0050 | 0.0253 *** |
|  | (−2.2880) | (−0.8995) | (−2.0483) | (3.6789) | (−0.8929) | (4.8598) |
| Constant | −13.6808 ** | 1.7498 | 0.0257 | 0.0117 | 0.0065 | 0.0362 |
|  | (−2.3933) | (0.3356) | (1.1103) | (0.6591) | (0.2773) | (1.6422) |
| Industry fixed effects | YES | YES | YES | YES | YES | YES |
| Province fixed effects | YES | YES | YES | YES | YES | YES |
| Time fixed effects | YES | YES | YES | YES | YES | YES |
| N | 1108 | 1089 | 1321 | 855 | 1288 | 888 |
| $R^2$ | 0.299 | 0.209 | 0.563 | 0.672 | 0.564 | 0.713 |

Note: The values in parentheses are t-values, and *, **, and *** indicate that the statistics are significant at the 10%, 5%, and 1% levels, respectively.

Furthermore, the heterogeneous impact of government–business relations on the green willingness enhancement effect of carbon trading pilot policies can be seen from the results in columns (3) to (6) of Table 6. Among them, the results of columns (3) and (4) show that only in areas where the closeness of government–business relations is high does the carbon emission trading pilot policy significantly enhance the innovation willingness of enterprises. Firstly, under high-closeness government–business relations, there is a smooth and effective information communication mechanism between the government and enterprises. The government can address policy doubts that enterprises may have in a timely manner, helping enterprises to fully understand the participation rules and management systems of the carbon trading market. At the same time, they can receive favorable signals from carbon trading policies and use supporting incentive policies more rationally, thus stimulating the green innovation vitality of enterprises. Secondly, the higher closeness of government–business relations implies that local government behavior acts more as a "supporting hand". Local governments actively solve difficulties encountered by pilot enterprises in the carbon trading market and their daily operations. This not only helps to enhance the sense of security for enterprises when participating in the carbon trading market but also helps enterprises to allocate more resources to green R&D investment and improve their tolerance for green innovation failures. This enhances their willingness to innovate. On the contrary, low closeness of government-business relations means that local governments lack attention to carbon trading enterprises. Local governments are likely to ignore the legitimate demands of enterprises in the carbon trading market and cannot quickly solve problems or protect the legitimate rights and interests of enterprises. This may reduce the enthusiasm and innovation ability of enterprises to participate in the carbon trading market, and it is not conducive to strengthening the motivation of enterprises' green innovation. At the same time, local governments lack communication with or among enterprises and fail to fully convey the policy orientation of carbon trading to enterprises. This is not conducive to deepening enterprises' understanding of these policies and awareness of green innovation, increasing the uncertainty faced by enterprises in participating in the carbon trading market and affecting their business decisions, which is not conducive to enhancing the willingness of enterprises to innovate.

Furthermore, columns (5) and (6) show that only in areas in which the integrity of government–business relations is high could the carbon emission trading pilot policy promote the innovation willingness of enterprises significantly. One possible explanation is that when there is high integrity in government–business relations, the relationship between the government and enterprises is more standardized and transparent. Local governments are able to guide the effective and reasonable allocation of resources in accordance with the goals of carbon trading policies to ensure that carbon emission trading pilot policies

can be better implemented. Additionally, when there is more openness and sincerity in the relationship between the government and enterprises, issues of corruption such as collusion between officials and businessmen, the abuse of power for personal gain, and power–money transactions can be rectified, and the space for power rent-seeking is reduced and eliminated. Thus, it is no longer feasible for enterprises to obtain special benefits including green subsidies and financing support from carbon trading policies or evade the penalties for violating carbon trading policies through interest transmissions. This situation encourages enterprises to avoid distractions and focus more on meeting the requirements of carbon trading policies, which makes it easier to identify and perceive the potential benefits and market opportunities of green innovation under carbon trading policies, thereby enhancing enterprises' willingness to innovate. In contrast, a low degree of integrity in government–business relations indicates that the government may engage in corrupt behavior and tend to "create rent" with their power. This can disrupt the orderly environment of the carbon trading market and hinder the effective implementation of related supporting incentive policies such as green subsidies. This situation forces enterprises to increase their operating and transaction costs and dampens their enthusiasm for innovation. Likewise, this abnormal government–business interaction negatively impacts normally operating enterprises, damages their trust in local governments, weakens the deterrence and incentive effects of carbon trading pilot policies on enterprises, and offsets the promotion effect of carbon trading pilot policies on enterprises' innovation willingness.

## 6. Friction in the Carbon Market's Impact on Corporate Green Innovation

This section empirically tests two types of friction: technical support and talent demand. In terms of technical support, this study employs the industry's green technology level for testing. Patents are regarded as the most direct reflection of the technical level of enterprises and reflect their technical innovation capabilities [51], whereas the proportion of green patents better characterizes the green technology bias and the direction of its progress [52]. Hence, following the practices of Dong and Wang [52], this paper employs the proportion of green patent applications made in each industry to represent the green technology level of each industry and splits the enterprise sample into two groups according to their industry median for estimation. Green patent data come from the State Intellectual Property Office. The empirical results are shown in columns (1) and (2) of Table 7. The PILOT×POST coefficient in column (1) is significantly positive at the 1% level, while the coefficient estimate in column (2) is not significant. Hence, the carbon trading pilot policy can only have a better green innovation incentive effect on enterprises in an industry with higher green technology levels. Thus, H4a is verified.

In terms of talent, following the practices of Quan et al. [36], this paper employs corporate human capital for testing and splits the enterprise sample into two groups, based on the median, for estimation. Human capital data come from the WIND database. The empirical results are shown in columns (3) and (4) of Table 7. The PILOT×POST coefficient in column (3) is significantly positive at the 1% level, while the coefficient estimate in column (4) is not significant. Therefore, the carbon trading pilot policy only has a better green innovation incentive effect on enterprises with higher human capital levels. Thus, H4b is verified.

**Table 7.** Friction test results of the carbon market's impact on enterprises' green innovation.

| | GreenPat | | | |
|---|---|---|---|---|
| Variables | High Level of Green Technology in the Industry (1) | Low Level of Green Technology in the Industry (2) | High Level of Human Capital (3) | Low Level of Human Capital (4) |
| PILOT×POST | 0.5273 *** | −0.0561 | 0.4076 *** | −0.0627 |
| | (4.7871) | (−0.6784) | (3.1492) | (−0.5505) |
| POST | −0.3070 * | −0.2622 ** | −0.4343 ** | −0.1607 |
| | (−1.9529) | (−2.1763) | (−2.5041) | (−1.0691) |

**Table 7.** *Cont.*

| Variables | GreenPat | | | |
| --- | --- | --- | --- | --- |
| | High Level of Green Technology in the Industry (1) | Low Level of Green Technology in the Industry (2) | High Level of Human Capital (3) | Low Level of Human Capital (4) |
| PILOT | 0.2091 * | 0.4143 *** | 0.4906 *** | 0.4503 *** |
| | (1.8328) | (4.7550) | (3.9706) | (3.5049) |
| SIZE | 0.3333 *** | 0.1823 *** | 0.2467 *** | 0.3195 *** |
| | (7.5115) | (4.2871) | (4.8152) | (6.0867) |
| AGE | 0.1000 | 0.1181 * | −0.0335 | 0.0707 |
| | (0.8995) | (1.6581) | (−0.3206) | (0.7119) |
| LEV | 0.5307 ** | 0.2373 | 1.1024 *** | 0.2535 |
| | (2.5391) | (1.4263) | (4.0156) | (1.0897) |
| ROA | 1.3645 ** | 0.2275 | −0.2219 | 0.6629 |
| | (2.0957) | (0.5960) | (−0.3522) | (0.9213) |
| CFO | 1.3537 *** | −0.3629 | −0.0992 | 0.4740 |
| | (2.9948) | (−1.2330) | (−0.2529) | (0.8275) |
| SHARE | −0.1913 | 0.3832 *** | 1.2667 *** | 0.3386 |
| | (−0.9490) | (2.8579) | (5.8325) | (1.1835) |
| COCEN | −1.3634 *** | −0.3788 ** | −1.0023 *** | −0.6587 ** |
| | (−5.5191) | (−2.2416) | (−4.2894) | (−2.5275) |
| Constant | −7.7655 *** | −4.8228 *** | −5.7942 *** | −6.9478 *** |
| | (−8.1985) | (−5.2710) | (−5.0450) | (−6.1711) |
| Industry fixed effects | YES | YES | YES | YES |
| Province fixed effects | YES | YES | YES | YES |
| Time fixed effects | YES | YES | YES | YES |
| N | 1072 | 1125 | 809 | 808 |
| $R^2$ | 0.519 | 0.421 | 0.598 | 0.535 |

Note: The values in parentheses are t-values, and *, **, and *** indicate that the statistics are significant at the 10%, 5%, and 1% levels, respectively.

## 7. Extended Analysis: The Carbon Trading Pilot Policy and Quality of Corporate Green Innovation

At present, the rapid development of new technologies and new industries is giving birth to a new round of the industrial revolution characterized by green, intelligent, and sustainable development. In 2021, the State Council's Guiding Opinions on Accelerating the Establishment and Improvement of a Green, Low-Carbon and Circular Development Economic System clearly stated that promoting high-quality development and high-level protection simultaneously, achieving the goals of carbon peak and carbon neutrality, will take China's green development to a new level. Improving the quality of green technology innovation is the only way to help China efficiently achieve its dual carbon goals, seize the huge opportunities in the global market of green technology and the green industry, and seize the high ground of global green development as well. Existing studies on carbon emission trading policies mainly focus on enterprises' R&D investment or the quantity of their green innovation output as the research variables, while neglecting the quality of their green innovation output (green patent quality) as a research factor.

Enterprises' green innovation output and quality are two indispensable dimensions for measuring enterprises' green innovation behavior comprehensively. Therefore, the relationship between carbon emission trading pilot policies and enterprises' green innovation quality deserves sufficient attention.

This paper further studies whether carbon emission trading pilot policies can improve enterprises' green innovation quality. Referring to the research of Liu et al. [53], this paper uses the number of citations of green patents applied for by enterprises within 2 years to measure the quality of green innovation (Citation). The green patent citation data come from the China Research Data Service Platform (CNRDS). The empirical results are shown in column (1) of Table 8. The PILOT×POST coefficient is not significant. This indicates that the carbon emission trading pilot policy has no significant impact on the green innovation quality of pilot enterprises. There may be two reasons for this. First, high-quality green innovation outputs often require a more substantial financial foundation and longer-term

asset investment. All seven carbon emission trading pilot areas have clearly stated that they support enterprises included in carbon trading policies applying for green credit and other financing services first. However, at present, green credit and green financial services for the carbon trading market are not perfect, and financial product innovation for the carbon trading market is lacking. Second, compared with the EU carbon trading market, the carbon price level in China's seven carbon trading markets is relatively low. This may not effectively drive enterprises to pursue high-quality green innovation; that is, they lack effective incentives to improve the quality of corporate green innovation activities. Finally, this article also analyzes how carbon trading pilot policies affect the quality of green innovation in enterprises differently based on subsamples of different industries with varying green technology levels and enterprises with varying human capital, and thereby reflect the frictions caused by technical support and talent constraints. The empirical results for the industry's green technology level are presented in columns (2) and (3) of Table 8. The PILOT×POST coefficient in column (2) is significantly positive at the 1% level, while the coefficient estimate in column (3) is not significant. This indicates that carbon trading pilot policies can effectively play a role in incentivizing the quality of green innovation in enterprises with higher industry green technology levels. The empirical results for enterprises' human capital are presented in columns (4) and (5) of Table 8. The PILOT×POST coefficient in column (4) is significantly positive at the 10% level, while the coefficient estimate in column (5) is not significant. This indicates that carbon trading pilot policies can effectively play a role in incentivizing the quality of green innovation in enterprises with higher human capital levels.

**Table 8.** CETS pilot policy and the quality of enterprises' green innovation.

| Variables | | Citation | | | |
|---|---|---|---|---|---|
| | Full Example | High Level of Green Technology in the Industry | Low Level of Green Technology in the Industry | High Level of Human Capital | Low Level of Human Capital |
| | (1) | (2) | (3) | (4) | (5) |
| PILOT×POST | 0.0692 | 0.3742 *** | −0.0897 | 0.2581 * | 0.0195 |
| | (0.9753) | (3.3722) | (−1.0464) | (1.7522) | (0.1808) |
| POST | −0.3666 *** | −0.4126 *** | −0.3954 *** | −0.4804 ** | −0.4535 *** |
| | (−3.8858) | (−2.6877) | (−3.1304) | (−2.3670) | (−3.6150) |
| PILOT | 0.4551 *** | 0.2825 ** | 0.4928 *** | 0.5972 *** | 0.3206 *** |
| | (6.3850) | (2.4370) | (5.3193) | (4.5073) | (2.6677) |
| SIZE | 0.3183 *** | 0.3736 *** | 0.2438 *** | 0.3026 *** | 0.3302 *** |
| | (9.9631) | (8.4052) | (4.8256) | (5.2618) | (5.8974) |
| AGE | 0.1557 *** | 0.2377 ** | 0.1449 ** | 0.0170 | 0.1934 ** |
| | (2.7741) | (2.1272) | (2.1958) | (0.1486) | (2.1050) |
| LEV | 0.1457 | 0.2261 | 0.0262 | 0.8856 *** | 0.1787 |
| | (1.1094) | (1.0885) | (0.1502) | (3.1153) | (0.7655) |
| ROA | 0.6672 ** | 1.6252 *** | 0.2246 | 0.4450 | 0.3200 |
| | (2.0084) | (2.7017) | (0.5492) | (0.7324) | (0.4538) |
| CFO | 0.2573 | 0.9561 ** | −0.1926 | 0.2366 | 0.7804 |
| | (0.9137) | (2.0869) | (−0.5836) | (0.5378) | (1.3012) |
| SHARE | 0.2729 ** | 0.0480 | 0.2999 ** | 1.1520 *** | 0.6144 ** |
| | (2.4147) | (0.2482) | (2.0868) | (4.4749) | (2.3287) |
| COCEN | −1.0294 *** | −1.5671 *** | −0.6350 *** | −1.4628 *** | −0.8332 *** |
| | (−7.2859) | (−6.1931) | (−3.5357) | (−5.8107) | (−3.3519) |
| Constant | −7.7889 *** | −9.0333 *** | −6.2279 *** | −7.0280 *** | −7.3526 *** |
| | (−10.9565) | (−9.2576) | (−5.6821) | (−5.5870) | (−6.3534) |
| Industry fixed effects | YES | YES | YES | YES | YES |
| Province fixed effects | YES | YES | YES | YES | YES |
| Time fixed effects | YES | YES | YES | YES | YES |
| N | 2197 | 1072 | 1125 | 809 | 808 |
| $R^2$ | 0.378 | 0.471 | 0.329 | 0.542 | 0.458 |

Note: The values in parentheses are t-values, and *, **, and *** indicate that the statistics are significant at the 10%, 5%, and 1% levels, respectively.

## 8. Conclusions and Suggestions

In the process of achieving a vision of "carbon neutrality", carbon emission trading is a crucial component that is indispensable and mainly promotes the green transformation of enterprises through market-oriented mechanisms. Using the PSM-DID model, this article empirically tests the impact of the carbon emission trading pilot policy that was launched in 2013 on green innovation in enterprises. The research found that, first, carbon emission trading policies can significantly promote the green innovation of enterprises: the greater the punishment and the higher the carbon price are, the more obvious the improvement of green innovation among enterprises is. Secondly, from the perspective of their paths of action, carbon emission trading policies can promote green innovation in enterprises by providing innovation resources and enhancing the willingness of enterprises to innovate, without imposing significant environmental costs on enterprises. Further analysis found that carbon trading policies can effectively play a role in providing innovation resources only in areas with relatively low local government competition and that amicable and transparent government–business relations can enhance the positive impact of carbon trading policies on the innovation willingness of enterprises. Thirdly, considering the friction in the process of carbon trading's promotion of green innovation in enterprises, the level of green technology in the industry and the human capital of enterprises may reduce the incentive effect of carbon trading on green innovation in enterprises. Fourthly, currently, carbon trading policies have no significant impact on the quality of green innovation in enterprises, and the level of green technology in the industry and the human capital of enterprises may also limit the full potential of carbon trading's incentive effect on the quality of green innovation among enterprises. This article expands on previous research on the impact of carbon emissions trading in China on the number of corporate patents submitted [7] and clarifies the path of action of the pilot policy of carbon trading on corporate green innovation from the perspective of corporate innovation resources and innovation willingness. This paper echoes the research conclusions of Song et al. [16] and Yu et al. [54], that the pilot policy of carbon emission trading has promoted the green innovation of enterprises. The significant difference between the conclusions of this paper and those of Zhang et al. [11] and Chen et al. [2] is mainly due to the differences in time intervals and measurement methods used.

First, the sampled research interval of this paper is 2009–2019. When selecting the sampled research interval, this paper avoids the impact of the 2008 financial crisis on corporate operations and includes the important event of the National Development and Reform Commission issuing the "National Carbon Emission Trading Market Construction Plan (Electricity Industry)" in 2017, which includes the overall design of the national carbon trading market and its lagging effects.

Second, in terms of the measurement method for green innovation, this paper believes that, compared with utility model patents, invention patents often contain more independent intellectual property rights, emphasize breakthroughs and novelty, and better reflect an enterprise's pursuit of "quality" in innovation.

At the same time, this article's research shows that the implementation of carbon emissions trading policies may encounter the following challenges and obstacles:

Firstly, the impact of carbon prices on the quality of corporate green innovation is minimal, which is not conducive to stimulating enterprises to carry out high-quality green innovation. The reason is that the overall level of carbon prices in various pilot areas is currently low; they cannot effectively play the role of carbon asset value signals, cannot effectively mobilize enthusiasm for corporate green innovation, and cannot put enough pressure on high-carbon enterprises, so they cannot stimulate high-quality green innovation in enterprises.

Secondly, the carbon trading pilot policy has not forced enterprises to improve the quality of their green innovation by increasing corporate environmental costs. This may be because the current carbon emissions trading pilot policy only imposes fines and cancels fiscal subsidies and other related punishment systems for non-compliant emission control

enterprises, so the cost pressure directly imposed on enterprises by the pilot policy is low and cannot prompt enterprises to increase their environmental costs.

In addition, in enterprises with lower levels of industry green technology and human capital, it is difficult for the carbon emissions trading pilot policy to effectively play the role of a green innovation incentive. The reason is that in industries with lower levels of green technology, the atmosphere of green innovation is not strong, and it is difficult to trigger imitation pressure for pilot enterprises, while the lower level of human capital is not conducive to enterprises learning and digesting carbon trading rules and breaking through the difficult bottlenecks encountered in green innovation.

Finally, in areas where local government competition is fierce and the degree of integrity and closeness of government–business relations is low, the promotion effect of carbon emissions trading pilot policies on corporate R&D investment is also limited. This may be because the higher local government competition pressure and the lower degree of closeness of government–business relations mean that local governments lack attention to or communicate with the enterprises included in carbon trading, neglecting the demand for government support in the process of enterprise carbon market trading, which is not conducive to deepening enterprises' understanding of the policies and green innovation and may reduce the enthusiasm and innovation ability of enterprises to participate in the carbon trading market, which is not conducive to enterprises increasing their R&D investment.

Based on the conclusions of this article, our policy recommendations are as follows:

First, optimize the internal design of the carbon emission trading policy to promote high-quality green innovation in enterprises. According to the research conclusions of this article, the greater the punishment and the higher the carbon price are, the more they can promote green technology transformation and upgrading in enterprises. Therefore, we should draw lessons from the design of default penalty systems in different carbon markets, appropriately increase the penalties for defaulting enterprises and enrich the penalty mechanisms, such as establishing a default "blacklist" system and enhancing the penalties for fraudulent carbon emission data behavior, to ensure the efficient and orderly operation of the carbon trading market. In addition, the supply of quotas in the carbon trading market exceeds the demand, resulting in low carbon prices, which is not conducive to achieving the goal of "carbon neutrality". Therefore, the government should reduce the issuance of quotas gradually and appropriately to increase carbon prices and reflect the value of carbon assets.

Second, improve the rules that support carbon emission trading to encourage enterprises to enhance the quality of their green innovation. This study found that carbon emission trading policy can promote green innovation in enterprises by providing innovation resources and enhancing the innovation willingness of enterprises. At present, the system that the carbon emission trading management measures stipulate is too principled and framework-oriented, lacking operability. The pilot areas should avoid "sports-style carbon reduction"; actively formulate a series of supporting rules, such as specific financial support and green financial support; accelerate the innovation of green financial products, such as green bonds, green insurance, and green funds; and incorporate the quality of green innovation into the important standards that the relevant policies on carbon emission trading support. For example, establish a credit management system that matches the characteristics of green lenders, dynamically adjust their credit resources based on the green R&D activities and output quality of pilot enterprises, and ultimately achieve the legislative goal of sustainable development.

Third, help enterprises overcome the friction found in technical support and human capital. For industries with a low overall level of green technology adoption, the government needs to provide targeted technical guidance to help them solve problems related to green technology. At the same time, the government should encourage and guide the exchange of green technologies between different industries to encourage enterprises' sustainable innovation. In terms of talent supply, the government can, using scientific research platforms such as universities and research institutions, actively carry out knowledge

and skill training, which is, in multiple fields such as carbon asset management, green technology and, finance, improving the ability and competitiveness of green low-carbon talent and laying a talent foundation for the development of the carbon trading market as well as enterprises' green technology innovation.

Fourth, promote the construction of a collaborative and clean government–business relation system. Firstly, break away from the view of performance that focuses on GDP only, establish a comprehensive and scientific development outcomes assessment and evaluation system, incorporate environmental development indicators including carbon reduction and technological innovation development into it, and urge local governments to pay attention to green development. Secondly, further strengthen the construction of collaborative and clean government–business relations. On the one hand, improve the construction of government–enterprise communication mechanisms and build close and mutual political–business relationships. Setting up regular government–enterprise exchange meetings, providing door-to-door services, and organizing policy lectures and other methods to make the government–enterprise communication mechanism dynamic, normalized, and simplified will thereby effectively transmit the national environmental policy orientation as well as enhancing the enthusiasm of enterprises for green innovation. On the other hand, strengthen the supervision of the government's behavior and build a clear and responsible government–business relationship, by setting up a reporting and complaint platform, regularly visiting and researching enterprises, building an enterprise satisfaction evaluation system for local governments, and using other methods to actively resist problems such as the corruption and inaction of local government officials, to provide a suitable business environment for enterprises' green innovation as well as stimulate enterprises' enthusiasm for green innovation.

The research in this article has generated conclusions of certain value, but due to data limitations and the influence of other uncontrollable factors, this study still has certain limitations. Firstly, in terms of sample selection, due to the limitations of data acquisition, this article only uses A-share listed companies as its research sample and does not consider non-listed companies. Secondly, this article only measures the intensity of punishment based on the comprehensive score of the punishment system of each pilot and cannot accurately and comprehensively measure the differences in the punishment system of each pilot. Finally, this article only considers the heterogeneity of the industry's green technology level and corporate human capital. According to the strategic tripod method used, institutional conditions, the industry environment, and corporate resources all play a role in corporate strategy and performance, and it is very important to explore the joint effect of the three. It is expected that future research could conduct a deeper investigation of the carbon emissions trading policy in terms of the above aspects.

**Author Contributions:** Conceptualization, H.Y., Z.W., H.Z. and Q.W.; methodology, H.Y., Z.W., H.Z. and Q.W.; software, Z.W. and H.Z.; validation, Z.W. and H.Z.; formal analysis, H.Y., Z.W. and H.Z.; investigation, H.Z.; resources, H.Z. and Q.W.; data curation, Z.W. and H.Z.; writing—original draft, Z.W.; Writing—review & editing, H.Y.; supervision, H.Z. and Q.W.; project administration, H.Y. and Q.W.. visualization, H.Y., Z.W., H.Z.; supervision, H.Y. and H.Z.; project administration, H.Y.; funding acquisition, H.Y. All authors have read and agreed to the published version of the manuscript.

**Funding:** This research was funded by [Hunan Provincial Department of Science and Technology, Hunan Provincial Natural Science Foundation Surface Project, 'Research on the Impact Effect and Mechanism of Local Industrial Chain Policy on Improving the Level of Industrial Chain Modernization'] grant number [2023JJ30708]. [National Planning Office of Philosophy and Social Science, National Social Science Fund General Project, 'Research on the Policy Mechanism and Impact Effect of Technological Innovation of Key Strategic Material Enterprises'] grant number [19BJY039]. [Hunan Department of Education, Hunan Provincial Department of Education Think Tank Project, 'Research on New Material Technology Innovation Policy and Enterprise Innovation Breakthrough: Mechanism, Path and Effect'] grant number [19k098].

**Institutional Review Board Statement:** This study was conducted in accordance with the Declaration of Helsinki and approved by the Research Ethics Committee of TOHOKU UNIVERSITY (15 May 2023).

**Data Availability Statement:** Data is contained within the article.

**Conflicts of Interest:** The authors declare that they have no competing financial interests.

## Appendix A. Definition of "Intention"

The "Intention" defined in this paper refers to the tendency of enterprises to carry out innovative activities and make innovative investments in order to improve their level of innovation. The theory of planned behavior considers that willingness is the most direct predictive variable able to explain behavior. The intention of an enterprise to innovate is highly related to its innovative behavior. The stronger the intention of an enterprise to innovate, the more it invests in innovative activities. At the same time, an innovative investment is a direct response to the intention to innovate of an enterprise, so R&D investments can be used to measure the intention to innovate of an enterprise. Based on the above analysis, this paper refers to the practices of Jiang and Liu [43] and collects the data on the ratio of R&D investment to operating income of A-share listed companies in Shanghai and Shenzhen from 2009 to 2019 from the China CSMAR database and Wind database as a proxy variable for the willingness to innovate of enterprises. It is specifically expressed as Intention = R&D Investment / Operating Income.

## Appendix B. Method of Distinguishing between the High and Low Levels of Government Competition Relations and Government–Business Relations Groups

We draw on the research of Du Yu [47], who selects the level of economic catch-up as a proxy variable for government competition, where government competition = (highest GDP of neighboring cities / local GDP), i.e., the pursuit of surrounding areas with higher economic levels. Based on the median, our enterprise sample is divided into two groups, a high degree of government competition and a low degree of government competition, to estimate the green subsidy incentive effect of the carbon trading policy. The data on government competition come from the China Statistical Yearbook. Similarly, this paper uses the median division method and, based on the median of the closeness index and the integrity index, the enterprise sample is divided into two groups to estimate the effect of the carbon trading policy on the enhancement of innovation willingness. The measurement methods used for the closeness index and the integrity index refer to the city government–business relationship indicators constructed by Nie et al. [50].

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
