# Peer review of "How Does Low-Carbon Financial Policy Affect Corporate Green Innovation?—Re-Examination of Institutional Characteristics, Influence Mechanisms, and Local Government Behavior"

_sustainability, doi:10.3390/su16103944_

Round 1

Reviewer 1 Report

Comments and Suggestions for Authors

While the paper "How Does Low-Carbon Financial Policy Affect Corporate Green Innovation?—Re-Examination of Institutional Characteristics Influence Mechanism and Local Government Behavior" provides substantial contributions to the field, there are areas for improvement that could enhance its impact and clarity:

The paper integrates various theoretical perspectives, including the financial instrument characteristics of carbon trading and the strategic tripod theory. While this approach is comprehensive, it could benefit from a more explicit articulation of how these theories intersect and inform the study's hypotheses. 

The study focuses on China's CETS pilot policy and its impact on corporate green innovation within a specific timeframe. While this provides valuable insights, the paper could explore how the findings relate to broader contexts, such as comparisons with carbon trading systems in other countries or regions. 

Discussion of Policy Implementation Challenges: While the paper effectively analyzes the policy's impact on green innovation, it could further explore the challenges and obstacles encountered in the implementation of the CETS pilot policy. This includes administrative hurdles, compliance issues, and the role of market mechanisms in facilitating or hindering policy effectiveness. 

Addressing these points could enrich the paper's contribution to the literature, offering more nuanced insights and practical implications for policymakers, businesses, and researchers engaged in the transition towards a low-carbon economy.

Comments on the Quality of English Language

Minor editing of English language required

Reviewer 2 Report

Comments and Suggestions for Authors

The manuscript is written clearly. The scientific problem of assessing the effectiveness of low-carbon programs is important. The econometric methods used in the work are described in detail in Section 3.1 and they are quite adequate for assessing the effectiveness of low-carbon programs.

Empirical results presented in Sections 4-7 are quite convincing, although they seem predictable in advance. All conclusions are confirmed by rigorous statistical analysis.

Some minor remarks:

-              Line 560. Authors write “As shown in Figure 3, the hollow circles in the figure represent the size of the regression coefficients that depict the 561 effect of the carbon emission trading pilot policy”. All the circles are of the same size – is it correct?

-              Line 17 – typos

-              It seems to me that the literary review is slightly incomplete. The authors focus exclusively on the works of authors from China. However, recently a lot of interesting works have appeared on the same topic regarding the effectiveness of low-carbon policies in other countries. In particular, the work https://link.springer.com/article/10.1007/s13132-023-01658-2 analyze the efficiency of low-carbon finance in China, India, Brazil, and the USA.

Reviewer 3 Report

Comments and Suggestions for Authors

The paper is very interesting.

The topic is also very interestin.

The empirical results are adequately presented.

Some suggestions for possible improvement.

1. Section 2 (Hypotheses) need to be strenghthen. The empirical hypotheses should not be justified from on;lew few (in moste of them , fron 1 or 2 referrences to findings from other researh works). For example the first hypothesis seems to be justified only by 1 citation)..

2. In model structure, it is not very clear whwether the measures of Penalty are sectoral (that means the same number for many different firms) or in firm level (the penalty for each firm separately, which is more representative and different for each case).

3. A further explanation could be usefull on the purpose of the use of both carbon market and non-carbon market-included enterprises. And a discussion on possible differences in the findings section.

4. It is not specified if the sample comes from similar sectors or not, as many differences may appear in the adoption of innovative green technology, relative to the sector a firm belongs to.

5. In the cocnlusion section, a discussion with some referrence (that should be included also to the section 2 as already mentioned), for possible similarities or differences will be usefull for the academic soudness of the paper

Reviewer 4 Report

Comments and Suggestions for Authors

The paper is a theoretical and empirical (microeconometric) study of the effects of the carbon emissions trading system (CETS) on eco-friendly(green) innovation in China, covering the pilot years and regions of the program. The paper develops a simple theoretical framework, resulting in four groups of hypotheses (H1–H4), which are then tested with the help of the difference-in-difference (DID) model, including several robustness checks. The treatment and control groups of firms are matched using the propensity score matching method (PSM).

Overall, the concept of the paper and the methodological side seem sound. There are clearly defined goals, hypotheses, and methodology.

The results are well-presented and commented on. Still, some improvement is required before the paper can be published:

  1. It is recommended that the paper has a proper previous research section (currently, the core of it resides in the introduction) and/or a clear explanation of how it is divided among sections (some sources are also found in the theoretical analysis).
  2. The theoretical analysis requires references to research at the world level to be considered credible (currently, primarily publications from the PRC are cited, which does not make the resulting model "universal"; alternatively, it should be presented as a China-specific model).
  3. Details on the calculation of Intention and a selection of high/low levels of government competition and government-business relations groups need to be provided at least as an annex to the paper.
  4. The paper needs a proper discussion of the results with a comparison to previous research to explain its value added.

Lines 96–106 include repeated sentences.

Comments on the Quality of English Language

There are punction problems: spaces after full stops are missing.

Reviewer 5 Report

Comments and Suggestions for Authors

The research problem addressed is important and topical. It is not yet adequately explained in the literature. The decarbonisation of the global economy (including China's) is nowadays a huge challenge on a mega-, macro- and micro-economic scale. The use of financial decarbonisation tools by the state in different economic sectors and enterprises and their effectiveness in reducing CO2 emissions and in implementing green technologies is an important research problem in all countries - developed and developing. The study is devoted to this issue in relation to China, where for more than a decade the government has applied a special pilot policy to influence the decisions of companies buying CO2 emission allowances. The authors set out to investigate how the CETS pilot policy, introduced since 2013, affects the propensity of companies to create and implement green innovations. However, their research has a broader context. The research has identified the impact of local governments and their relationships with entrepreneurs, as well as the potential of companies (human capital, technological level and R&D spending) on the use of CETS to implement green innovations. This constitutes a major added value of the article.

Comments and recommendations to improve the article:

1. Introduction

(a) The Introduction lacked information about the aim(s) of the study. Information about the aim of the study was not supposed to be given by the authors until p. 8 (line 386). However, instead of the aim of the study, two further research hypotheses were stated: H4a and H4b. Thus, the research aim is not stated in the paper. Therefore, the purpose of the research/article should be completed in the Introduction.

(b) In this part of the paper, it is necessary to give general information about the research methods used, the data sources. This aspect is discussed in more detail later in the paper.

c) The authors should also write what the structure of the study is - what parts/sections it consists of.

2 Literature review

The literature review should be improved. It is very superficial. Too little content has been devoted by the authors to showing the state of research in the field of their research problem, to show what has been achieved so far and by whom. There is also a lack of assessment of the state of the art of the research problem not only in the Chinese area, but also in other countries in the world, e.g. Europe, EU, USA, Japan.

Examples of additional sources:

Teixidó, Jordi & Verde, Stefano F. & Nicolli, Francesco, 2019. The impact of the EU Emissions Trading System on low-carbon technological change: The empirical evidence, Ecological Economics, Elsevier, vol. 164(C), pages 1-1.

Fung K. C., Climate Change Policy, Emission Trading Schemes and Carbon Pricing: California and Tokyo, Journal of Economic and Public Finance, 2023, 1.

Guo Q., et al. Carbon emission trading policy, carbon finance and carbon emission reduction: evidence from a quasi-natural experiment in China, Economic Change and Restructuring, 2022, 55: 1445-1480.

3 Discussion of results

(a) In the section of the article where the authors discuss the research results obtained, there is a lack of reference to the results in this area by other authors. It is difficult to see whether these results are unique to China or similar to what has been studied for other countries. A comparison of these results would make it possible to determine whether the CETS pilot policy is effective, whether it should be improved, etc.

4 Conclusions

(a)    The final part of the article should point out the research limitations that have occurred and write something about the direction of future research.

Editorial comments requiring improvement:

(a) The text of the article sometimes lacks the publication dates of the cited sources.

(b) In the text, the authors have written the same text twice on p. 3 (compare lines: 96 to 101 and lines 101 to 107).

Round 2

Reviewer 5 Report

Comments and Suggestions for Authors

The corrections and additions made by the authors are appropriate and in accordance with the reviewer's orders.

Appendix a and Appendix B appeared at the end of the new version of the article, where the authors included additional definitions and explanations. This is a good thing, but in the text of the article the authors should include references to this information in the right places (cite the correct Appendix).

In addition, the authors should check the text from an editorial point of view. In a very large number of cases, it is noticeable that there is no space between the final words of the text and the cited source. Examples:

Verse 87-89:

As significant participants in the carbon trading market, enterprises are both major carbon emitters and core organisations for the development of low-carbon products(Zhao et al., 2021).

Verse 106:

Hu Jun et al.(2020) found ............

Editorial corrections are also needed in the list of bibliographies and in the notation of hypotheses (the authors used italics or the notation without italics).

With these minor additions and editorial corrections, the article can be published.

Author Response

Thank you for your second round comment. Based on your suggestions, we have made the following modifications:

  1. Corrected the issue where there was no space between the last word and the closing parenthesis when citing references.
  2. In the reference section, we now use italics to denote journal names.
  3. All font styles of hypotheses have been consistently changed to italics.
  4. Added links pointing to the appendices: "see details in Appendix A" and "see details in Appendix B".
  5. Additionally, we addressed other minor issues in the article, such as capitalization and spacing after periods.

We hope these revisions enhance the article’s consistency and readability. Once again, thank you for your comment!
